# Enhancing Performance of Reservoir Computing System Based on Coupled MEMS Resonators

**DOI:** 10.3390/s21092961

**Published:** 2021-04-23

**Authors:** Tianyi Zheng, Wuhao Yang, Jie Sun, Xingyin Xiong, Zheng Wang, Zhitian Li, Xudong Zou

**Affiliations:** 1The State Key Laboratory of Transducer Technology, Aerospace Information Research Institute, Chinese Academy of Sciences, Beijing 100010, China; zhengtianyi17@mails.ucas.ac.cn (T.Z.); sunjie17@mails.ucas.ac.cn (J.S.); xiongxy@aircas.ac.cn (X.X.); wangzheng02@aircas.ac.cn (Z.W.); lizt@aircas.ac.cn (Z.L.); zouxd@aircas.ac.cn (X.Z.); 2School of Electronic, Electrical and Communication Engineering, University of Chinese Academy of Sciences, Beijing 100010, China

**Keywords:** reservoir computing, coupled resonators, MEMS

## Abstract

Reservoir computing (RC) is an attractive paradigm of a recurrent neural network (RNN) architecture, owning to the ease of training and existing neuromorphic implementation. Its simulated performance matches other digital algorithms on a series of benchmarking tasks, such as prediction tasks and classification tasks. In this article, we propose a novel RC structure based on the coupled MEMS resonators with the enhanced dynamic richness to optimize the performance of the RC system both on the system level and data set level. Moreover, we first put forward that the dynamic richness of RC comprises linear dynamic richness and nonlinear dynamic richness, which can be enhanced by adding delayed feedbacks and nonlinear nodes, respectively. In order to set forth this point, we compare three typical RC structures, a single-nonlinearity RC structure with single-feedback, a single-nonlinearity RC structure with double-feedbacks, and the couple-nonlinearity RC structure with double-feedbacks. Specifically, four different tasks are enumerated to verify the performance of the three RC structures, and the results show the enhanced dynamic richness by adding delayed feedbacks and nonlinear nodes. These results prove that coupled MEMS resonators offer an interesting platform to implement a complex computing paradigm leveraging their rich dynamical features.

## 1. Introduction

Artificial neural networks (ANN) have played an important role in the current boom in artificial intelligence (AI), especially with the invention of the Internet of Things (IoT) and ubiquitous sensing [1]. ANN has strong self-learning adaptability, parallel information processing capabilities, and nonlinear mapping capabilities, but they require complex and time-consuming algorithms to train the connection weights in the large-scale network. When interested in processing time-dependent tasks, standard feed-forward ANNs can no longer meet the requirements, so the recurrent neural network (RNN) has gradually become the mainstream [2,3,4]. RNN requires complex algorithms to train the connection weights, however, which causes slow convergence and consumes lots of calculations [5,6]. Therefore, reservoir computing (RC) gradually attracted attention because the weights of its recursive network are initialized randomly and untrained.

Reservoir computing (RC) is a brain-inspired computational framework suited for temporal data processing, owing to its derivation from the recurrent neural network (RNN) [5,7]. The main difference between RC and conventional RNNs is that the weights on the recurrent connections in the reservoir are not trained, but only the weights in the readout are trained, which avoids the well-known limitations of RNN, such as iterative parameter optimization and the condition of convergence [8]. More importantly, the fixed reservoir without updating is suitable for hardware implementation using various nonlinear dynamical systems. The hardware reservoir needs two crucial properties, nonlinearity and fading memory, which can be realized by using numerous randomly interacting nonlinear nodes or using a time-delayed nonlinear system. Actually, RC based on the time-delayed nonlinear system greatly reduces the implementation difficulty, which has been successfully demonstrated in a wide variety of systems, such as memristor [9,10], electronic [8,11,12], spintronic [13,14,15], optoelectronic [16,17,18], all-optical [19,20,21,22], quantum [23], and mechanical resonator [24,25,26,27]. In spite of these encouraging results, the RC implementation based on the single nonlinear node with single time-delay feedback adopted in these previous works has a limited memory capacity. The memory capacity is a key property of the RC that allows the processing of dynamical signals. The RC structures with the multiple delayed feedback or bidirectionally coupled nodes was proven to increase the processing speed of the system by boosting the memory length of the RC system, which are discussed numerically in detail in Appeltant’s doctoral dissertation [28]. Furthermore, this pioneering work inspires many studies on different configurations for the time-delayed RC. Particularly, double optical feedbacks RC structures are proposed to increase prediction performance under the condition that the long delay time and short delay time meet the specific relationship [29]. In addition, the mutually delay-coupled semiconductor lasers structure was studied for enhancing calculation speed and accuracy of the RC systems [30,31,32]. Despite these instructive works based on the optoelectronic devices promoting research on RC with the multiple delayed feedbacks and coupled nonlinear nodes, the potential of RC with other nonlinear devices and the reason why the coupled RC structure improves the accuracy has not yet been fully investigated.

In this work, we present a couple-nonlinearity RC structure with double-feedbacks (CRSD) based on the coupled MEMS resonators with the enhanced dynamic richness to promote the performance of the RC. Moreover, we first put forward that the dynamic richness of RC comprises linear dynamic richness and nonlinear dynamic richness, which can be enhanced by adding delayed feedbacks and nonlinear nodes, respectively. In order to elaborate this point in detail, we compare three typical RC structures, a single-nonlinearity RC structure with a single-feedback (SRSS), a single-nonlinearity RC structure with double-feedbacks (SRSD), and the couple-nonlinearity RC structure with double-feedbacks (CRSD). In addition, we describe the evolution for the promotion of the performance of the three structures. Furthermore, four different tasks—the parity benchmark task, nonlinear autoregressive moving average (NARMA) task, isolated spoken digits recognition, and human action recognition—are investigated to prove the feasibility of these RC structures. The results show the enhanced dynamic richness by adding delayed feedbacks and nonlinear nodes.

The rest of this paper is organized as follows. In Section 2, we describe the methods we used in this paper, including the three typical structures of the RC system, the nonlinear mapping node of the RC structure, and the dynamic richness analysis of three RC structures. In Section 3, we give a detailed discussion about the numerical results of our system on four typical tasks. In Section 4, the paper ends with a brief conclusion and an outlook for possible future work.

## 2. Methods

### 2.1. The Structures of the Reservoir Computing

A general single nonlinear node RC structure with a single-feedback (SRSS) can be conceptually decomposed into three layers, which are shown schematically in Figure 1a. The input layer (in blue) is the preprocessing stage of the RC system. After transformed by the mask $m\left(t\right)$, $m\left(t+T\right)=m\left(t\right)$, the original signal $u\left(t\right)$, $nT\ll t<\left(n+1\right)T,$ with the duration T is injected into the reservoir layer. The mask is random samples at the range of (−1, 1), with zero mean and unit variance, which breaks the symmetry of the input data and creates various transient trajectories for the nonlinear nodes in a transient state [17]. Input gain β determines the amplitude of the input signal of reservoir layer, which determines the position of the system operating point. The reservoir layer (in orange) in Figure 1a consists of a single nonlinear node and a delayed feedback loop. Particularly, a nonlinear node is used to nonlinearly map the time-dependent input to a high-dimensional feature space based on its own nonlinear output capability and inherent attenuation characteristics, which are represented by the state of the nodes forming the reservoir. This nonlinear mapping makes the initial complex input linearly separable in the new space based on the reservoir state, so a simple linear network layer (output layer) can be used for further software processing [33]. The delayed feedback loop helps the RC system build long-term memory [34]. This property assures the execution of computations that demand input retrieval for several time steps in the immediate past so that the system can handle prediction tasks that need more memory length. N together with the duration θ, θ=T/N, are denoted as virtual nodes because they exist in a time-multiplexed way rather than the nodes corresponding to real physical spatially distributed and interconnected nodes. The values of virtual nodes $x_{i}$ are injected into the output layer (in green) for the postprocessing; $x_{i}$ are linearly combined, $y\left(n\right)=\overset{N-1}{\underset{i=0}{\sum }}Wx_{i}\left(n\right)$, where W is a set of readout weights that are trained by the ridge regression algorithm.

The dynamic reservoir layer is the key to determine the performance of the RC because of its ability to nonlinearly transform input data into high-dimensional space. In addition, the reservoir is fixed and only the readout is trained with a simple method such as linear regression and classification. Therefore the reservoir is sensitive to its optimizable architecture properties, and attention should be paid to the optimization for the reservoir layer rather than the optimization of the input and output layers. The dynamic richness describes the response of the reservoir, which provides memory and nonlinear computing capabilities of the system. Based on our analysis, the dynamic richness consists of linear richness and nonlinear richness. The linear richness represents the linear combination of the current virtual node state and various previous nodes information, which are determined by the delay feedback loop. The general SRSS has only one delay feedback loop with a delay feedback gain α and a delay length τ. Here, we define $\tau =\left(N+1\right)T/N$, where T is the period of the input data, making the system in an unsynchronized state where the virtual node is connected with several dependent nodes in a short period of time [17]. Thus, the reservoir of SRSS has only one input channel and one output channel. To enhance the linear richness of the RC system, we add another feedback loop, as shown in Figure 1b. Compared with SRSS, SRSD has an additional delay feedback loop. The short delay feedback loop (in orange) has the same delay length as SRSS’s, $\tau_{1}=\tau$, and the delay length of the long delay feedback loop (in black) is $\tau_{2}=2\tau$. The short feedback gain equals the long feedback gain, $\alpha_{1}=\alpha_{2}$. In the output layer, all responses from the virtual nodes of the short delay feedback loop are weighted and linearly summed. This structure is regarded as a transition structure for CRSD and has one input channel and two output channels. Since the nonlinear richness represents the nonlinear mapping ability of the nonlinear unit itself, we change the nonlinear mapping node from a single node to the coupled nonlinear node based on SRSD to optimize the nonlinear richness of the RC system. As illustrated in Figure 1c, CRSD is a system with a coupled nonlinear unit as the core, and two output channels and one input channel. The parameters of two delay feedback loops (in orange and black) are the same as those in SRSD. The details of the coupled nonlinear node are shown in the next section. Consequently, compared with the general structure SRSS, the CRSD increases the linear richness and nonlinear richness of the system.

### 2.2. The Nonlinear Mapping Node of the RC Structure

The nonlinear transforming of injected time-dependent data into a higher dimensional state is one of the most crucial properties of RC [35]. This transforming property determined by the nonlinear mapping node of the system is directly related to the accuracy of testing different data sets. The nonlinear mapping nodes of the SRSS and SRSD are based on the Duffing oscillator:(1)$$m\ddot{x}+\dot{cx}+k_{1}x+k_{3}x^{3}=Fcos\left(\omega t\right)$$
where x is the displacement of silicon beam, m is the lumped effective mass, c is the damping coefficient of the system, $k_{1}$ is the linear spring stiffness, $k_{3}$ is the nonlinear spring stiffness that affects the nonlinear behavior of the beam, ω is drive frequency, and F is harmonic force using the electrostatic drive. This higher-order ordinary differential equation only has one parameter ($k_{3}$) related to nonlinearity; that is, the nonlinear transforming property is not rich enough.

In this paper, we propose the coupled nonlinear mapping nodes to enhance the transforming property of CRSD, as illustrated in Figure 2. There are two resonators, Resonator 1 on the left and Resonator 2 on the right, enclosed by the red dotted line [36]. For subsequent silicon on glass (SOG) processing in our future work, each clamped-clamped beam is designed to be 40 μm thick, 500 μm long, and 6 μm wide, with a gap of 2 μm between two beams. Resonator 1 is driven and sensed using capacitive transduction while another vibrates simultaneously because of electrostatic coupling. The actuation is achieved using parallel plates of equal dimensions with 400 μm in length and 40 μm in thickness. The gaps of electrostatic drive and electrostatic detection are both 2 μm. The dynamics of the system is approximated by these equations:(2)$$m{\ddot{x}}_{1}+c{\dot{x}}_{1}+k_{1}\left(1+k_{3}x_{1}{}^{2}\right)x_{1}+k_{c}\left[1+k_{c3}{\left(x_{1}-x_{2}\right)}^{2}\right]\left(x_{1}-x_{2}\right)=Fcos\left(\omega t\right),$$
(3)$$m{\ddot{x}}_{2}+c{\dot{x}}_{2}+k_{1}\left(1+k_{3}x_{2}{}^{2}\right)x_{2}+k_{c}\left[1+k_{c3}{\left(x_{2}-x_{1}\right)}^{2}\right]\left(x_{2}-x_{1}\right)=0,$$
where $x_{1}$ and $x_{2}$ is the displacement of Resonator 1 and Resonators 2, respectively, m is lumped effective mass, c is the damping coefficient of system, $k_{1}$ is linear spring stiffness, $k_{3}$ is nonlinear spring stiffness, ω is drive frequency, F is harmonic force using the electrostatic drive, $k_{c}$ is the linear coupled stiffness, and $k_{c3}$ is the nonlinear coupled stiffness. These differential equations have two parameters ($k_{3}$, $k_{c3}$) to impact the nonlinearity of the system. Due to the coupled stiffness ($k_{c}$, $k_{c3}$), the nonlinear relationship between output and input is greatly strengthened. The simulated values are $k_{1}=586.6\;N/m$, $k_{3}=3.404\times 10^{12}\;N/m^{3}$, and $k_{c3}=5.21\times 10^{12}\;N/m^{3}$. When two resonators separated by an electrically coupled gap are subjected to different DC polarization voltage, the displacement component of the electrostatic force generated between the resonators forms a negative electrostatic spring, whose magnitude is [37]:(4)$$k_{c}=-\frac{\partial F}{\partial x}=-\frac{\varepsilon_{0}A}{g^{3}}\left(\Delta V_{b}^{2}\right),$$
where A is the coupled electrode area, g is the gap between two coupled electrodes, and $V_{b}$ is the DC polarization voltage, which is also shown in Figure 2. When $V_{b}=5\;V$, the linear coupled stiffness $k_{c}=-0.2878\;N/m$.

After processed by mask signal and input gain, and modulated by the drive voltage, the input signal is injected into the drive electrode of the coupled resonators. The coupled resonators produce two different current output signals. The current signals are converted to voltage signals by the custom transimpedance amplifier (TIA) and then through the band pass filter to remove the noise. The amplitude modulation signals restore the amplitude information through the envelope (ENV) and then enter the long delay feedback loop and short feedback loop. Finally, two feedback signals are injected into the input signal to form a closed loop. The values of the virtual node from the short feedback loop are used as the processing data in the postprocessing algorithm.

### 2.3. Dynamic Richness Analysis of Three RC Structure

#### 2.3.1. The Linear Richness of Dynamic Richness

The memory of the previous input is very significant in the RC system. Insufficient memory can degrade performance, thereby depriving the reservoir of all prediction or processing ability [28]. Linear richness, which depends on the delay feedback loop and represents the memory ability of the system, is a part of dynamic richness. After going through the delay feedback loop with delay length τ, the value of the virtual node $x_{i}\left(t\right)$ at time t inject into the input end and linear combine with the virtual node $x_{i}\left(t+\tau \right)$ at time t+τ. Compared with SRSS, the SRSD has an extra delay feedback loop to enhance the linear richness of the RC system.

Figure 3 shows the output signals of SRSS and SRSD with the same input spike. A voltage pulse of 0.6 ms duration is sent to the reservoir layer, as shown in the top panel. The middle panel shows the output amplitude signal of SRSS while the bottom presents the information from SRSD. Due to the intrinsic attenuation characteristics of the resonators, the output signal at each interval decays to zero after a few microseconds. Since SRSS has only one feedback loop, the output signal can maintain about five intervals. When the interval is greater than five, the information of the input pulse has already disappeared. Due to the addition of a long delay feedback loop, the output signal of SRSD can keep at least nine intervals, which indicates that SRSD can preserve the useful information of the input signal for a longer time.

In order to study the effect of the delay feedback loop on memory ability in more detail, we used memory capacity (MC) to make a quantitative analysis. Jaeger proposed a test for system memory capacity in 2002 [38]. The input $u\left(k\right)$ of the reservoir is the point derived from the uniform distribution in the interval (−0.8, 0.8). The output $y\left(k\right)$ is constructed as an infinite number of output sequences $y_{i}$, each output sequence is a copy of the time sequence u, which is shifted by i steps, so $y_{i}\left(k\right)$ is the construction of $u\left(k-i\right)$ for i=1⋯∞. In the actual test, the maximum value of i is selected high enough that it will not have a significant impact on the results. Therefore, we choose i=40 for efficiency. The total memory capacity is defined as the normalized correlation between the approximate value of the target returned by the readout layer and its associated delay input, expressed by the following formula:(5)$$u_{c}=\sum_{i=1}^{\infty }m_{i},$$
with $m_{i}$ being the memory function, that is to say, the normalized correlation between $\hat{y_{i}}$ and $y_{i}$, given by:(6)$$m_{i}=corr\left[y_{i}\left(k\right),\hat{y_{i}}\left(k\right)\right]=corr\left[y_{i}\left(k\right),u\left(k-i\right)\right].$$

Figure 4 shows the memory function as the function of the delay step of two RC structures. The blue line stands for the SRSS. The memory function decreases sharply with the increase of delay step. When the delay step increases to 10, the value of the memory function decays to near 0. That means this SRSS is not well suited for long-term prediction tasks. The red line represents the SRSD. When the delay step is less than seven, the memory function almost has no attenuation. When the range of the delay step is 7~28, the memory function decreases gradually. The MC of SRSD is 13.3412, which is more than twice that of SRSS, whose MC equals 6.3689.

Obviously, SRSD with two delay feedback loops has better memory capacity than SRSS with only one delay feedback loop. That is to say, due to the addition of a delayed feedback loop, the RC system enhances the linear richness. Since the coupled resonators have two output signals, we do not add more (three or more) feedback loops in the simulation.

#### 2.3.2. The Nonlinear Richness of Dynamic Richness

As noted above, the reservoir has the ability to nonlinear transform the time-dependent input signal into a higher dimensional state space. The higher dimensional input signal greatly simplifies the difficulty of linear regression in postprocessing. Compared with SRSD, the CRSD uses the coupled nonlinear resonators to replace the single resonators, which improves the nonlinear richness of the RC system. The coupled resonators have two output signals from each resonator, and they connect a long delay feedback loop and a short delay feedback loop, which is the same as SRSD.

Figure 5 shows the output signal of two RC structures without the delay feedback loop. The top panel shows the input square wave signal with a period of 0.2 ms. One of the crucial properties of the RC system is consistency, which means the same response output can be observed by using a repeated input signal [39]. The outputs of the middle panel and the bottom panel remain consistent in each period, which indicated that these two systems all have the consistency property. In order to facilitate comparison, the two output signals have been processed by the same normalization method. Compared with the output of SRSS, the output of CRSD transforms the input signal into a relatively more complex data stream. That means the same input represents more data with different values.

Because the value of the virtual node directly determines the quality of the classification, analyzing the value of the virtual node can more intuitively show the nonlinear mapping ability of the RC system. Figure 6 shows the histogram of the virtual nodes in three RC structures with the same input data. We randomly intercepted 50,000 points in the values of a virtual node output for histogram statistics. In order to facilitate subsequent comparisons, the values of the virtual nodes are all normalized. The virtual nodes of SRSS are shown in the top panel. Most virtual nodes are concentrated near (0, 0.1), and there are around 8000 points in the range of (0.12, 0.13). The middle panel shows the virtual node of SRSD. After adding another delay feedback, the distribution of the virtual node is relatively scattered. Most virtual nodes are concentrated near (0.1, 0.3), and there are around 5000 points in the range of (0.17, 0.18). Because of the coupled nonlinear resonators, the distribution of virtual nodes is very scattered, as shown in the bottom panel. Most virtual nodes are concentrated near (0, 0.6) and the range with the maximum number of points is also limited to around 2500. The value distribution of the virtual nodes is more uniform, indicating that the virtual nodes have more abundant states, and the abundant node states can greatly improve the performance of the reservoir system.

In brief, CRSD has a richer node status than the other two systems. That is to say, the system enhances its nonlinear richness because of the coupled nonlinear resonators. Compared with the general structure SRSS, the CRSD increases the linear richness through another delay feedback loop and nonlinear richness through the coupled MEME resonators, which we call dynamic richness of the system.

## 3. Results and Discussion

The performance of the system lies in the following two aspects, prediction ability and classification ability, which are assessed by four typical data sets to prove that the coupled nonlinear resonators improve the dynamic richness of the RC system. The prediction ability is numerically investigated via the parity benchmark prediction task and NARMA task, while the classification ability is discussed via the TI-46 isolated word task and human action recognition task.

### 3.1. Parity Benchmark Prediction Task

The prediction ability of the system is first assessed by means of the parity benchmark task. The parity function is considered as a benchmark, as it requires both memory ability and nonlinear computational capability [40]. For this task, $u\left(t\right)$ is a binary time-dependent signal that randomly switches between two states −1 and +1, where τ is its switching period. The nth-order parity function can be defined as:(7)$$P_{n}=\prod_{i=1}^{n}u\left(t-\left(i-1\right)\tau \right),$$
where n is the order of benchmark, τ is the period, and $P_{n}$ is the training target of this $n^{th}$-order parity function. As n increases, the difficulty of this prediction task will gradually increase. When n=1, the benchmark performs a linearly separable data set and easy to predict [41]. As n>1, the data of $P_{n}$ at this time are related to the previous n data. Namely, the system needs to have at least a nτ memory length to process this data set. One hundred fifty training data and 50 testing data are used for our test and we choose the parameters, θ=0.1ms, N=400, f=352,500 Hz, β=1, and $V_{dc}=20\;V$, $V_{ac}=2\;V$ for the better success rate. The success rate of $P_{2\sim 4}=100\%$, $P_{5}=95\%$, and $P_{6}=88\%$ is reported in [24]. The method of parameter selection has been explained in our previously published articles [42] and will not be explained here.

Figure 7 shows the performance of SRSS and SRSD when the order n equals 7. Because the success rate of $P_{1}\sim P_{6}$ has already achieved 100% for two RC systems, we do not show the result of $P_{1}\sim P_{6}$ here. The parameters of these two RC systems are the same except for the delay feedback gain. The delay feedback gain α=5 in SRSS while $\alpha_{1}=2.5$ and $\alpha_{2}=2.5$ in SRSD. The black dotted line represents the 50 target value of $P_{7}$. Compared with the result of SRSS (red line), the result of SRSD (blue line) is closer to the target value. At the same time, the success rate is 78% for SRSS and 100% for SRSD. Since the success rate has already up to 100% for n=1,2,3,4,5,6,7 with SRSD, there is no room for improvement, so we did not use CRSD to further optimize this data set.

The result of this prediction task shows SRSD with another delay feedback loop can improve the success rate of the RC system by increasing its linear richness.

### 3.2. NARMA Prediction Task

The nonlinear autoregressive moving average (NARMA) prediction task is one of the most popular tasks in the RC community and it is also a more difficult task than the parity benchmark. In this prediction task, the RC system is trained to predict the behaviors of the system, for instance, a nonlinear autoregressive moving average (NARMA) of order m driven by white noise. The mth order NARMA task is given by the following recursive formula:(8)$$y\left(n+1\right)=0.3y\left(n\right)+0.05y\left(n\right)\left(\sum_{i=0}^{m-1}y\left(n-1\right)\right)+1.5u\left(n-m+1\right)u\left(n\right)+0.1,$$
where $y\left(n\right)$ is the output target of the system, m is the order of this benchmark, and $u\left(t\right)$ stands for the white noise, which is the random input derived from a uniform distribution over the interval (0, 0.5). In this task, the RC system is trained in a sequence of 1000 data length and tested in the following sequence of 1000 data length. The performance metric used to evaluate NARMA is normalized mean square error (NMSE), $\mathrm{NMSE}=\frac{1}{L}\overset{L}{\underset{n=1}{\sum }}{\left(y\left(n\right)-\hat{y}\left(n\right)\right)}^{2}/var\left(y\right)$, where $y\left(n\right)$ is the target and $\hat{y}\left(n\right)$ is the prediction result of the system. We use three RC systems (SRSS, SRSD, and CRSD) to test the performance of this data set, and the same parameters were set as N=50, β=3.6, and $V_{dc}=80V$, and $V_{ac}=2\;V$. In particular, θ exhibits a crucial role in this task and we set θ=0.01 ms to enhance the connection of current input to the previous virtual node state. The prediction difficulty of the system increases exponentially with the increase of order m. We test the prediction accuracy of the three RC systems under the conditions of m=1,2,5, and 10.

Figure 8 shows the performance of three RC systems for the NARMA prediction test. The lines with different colors represent the different orders of prediction. The driving frequency f=352,000 Hz in SRSS and SRSD, while f=264,300 Hz in CRSD. The delay feedback loop α=0.6 in SRSS, while $\alpha_{1}=0.4$, $\alpha_{2}=0.4$ in SRSD and CRSD with two delay feedback loops. In the same structure, the NMSE increases gradually with the increase of order because of increasing difficulty. Under the condition of the same order m, the NMSE decreases as the structure is optimized from SRSS to SRSD to CRSD at the same time. When m=1, CRSD reduces the NMSE of the system by around 70% from SRSS, and when m=2,5, CRSD reduces the NMSE of the system by around 50%. Concurrently, CRSD reduces the NMSE of the system by around 35%.

The results of this complicated prediction task show CRSD with two delay feedback loops and the coupled resonators can decrease NMSE by increasing its linear richness and nonlinear richness.

### 3.3. TI-46 Isolated Word Classification Task

Through the isolated spoken word task, the classification performance of reservoir computing when inputting complex data is evaluated. This task is the classification of isolated audio sequences from a subset of the National Institute of Standards and Technology Texas Instrument-46 Corpus (NIST TI-46 Corpus) [43]. The data set has 500 audio sequences, each audio sequence represents a number (0–9), recorded ten times by five different female speakers. Five hundred audio waveforms with different time lengths are sampled at a rate of 12.5 kHz. Every input representing a spoken digit is preprocessed using the Lyon cochlear ear model [44] before injecting into the RC system. In the training phase, ten weight vectors are calculated for the ten numbers in the vocabulary, and then the winner-takes-all method is applied to select the actual numbers. Each classifier is trained to output a value of 1 when presented with an utterance of the digit corresponding to its category, and 0 otherwise. The highest averaged classifier corresponds to the correct digit. The performance indicator used to evaluate the isolated word classification is the word error rate (WER), namely, the fraction of digits incorrectly classified. There are only 500 sequences in the subset corpus, so we divide them into 10 parts and use the tenfold cross-validation method to avoid the influence of the specific division of available data in some processes (such as training and testing), so as to make the results more convincing. Ten sections were randomly selected, including nine for training and one for testing.

In this task, we use three RC systems (SRSS, SRSD, and CRSD) to test the performance, and the same parameters were set as N=100, θ=0.1 ms, β=200, and $V_{dc}=20\;V$, $V_{ac}=2\;V$. The driving frequency f=349,200 Hz is in SRSS and SRSD, while f=244,300 Hz in CRSD; the delay feedback loop is α=0.4 in SRSS, while $\alpha_{1}=0.2$, $\alpha_{2}=0.2$ in SRSD and CRSD with two delay feedback loops. The original result is WER = 0.2% in SRSS. After increasing the linear richness of the system, the result is improved to WER = 0.12% in SRSD. CRSD improves the dynamic richness of the system and the result is optimized to WER = 0.04%.

In order to visualize how the time-dependent data separation occurs and understand the recognition capabilities of the different RC structures, the t-distributed stochastic neighbor embedding (t-SNE) technique [45] is used to represent input data of the TI-46 data set in a 2D figure. The t-SNE technique is a nonlinear dimensionality reduction technique used to map high-dimensional data to a low-dimensional space with two dimensions or three dimensions. During data reduction, the probability of two vectors becoming neighbors is preserved, so that the structure in the data can be visualized. Figure 9 shows the t-SNE result of two structures. For all of the data points of the digit isolated word, each number is represented by a colored dot. Figure 9a shows the result of the TI-46 raw data only after processing by the Lyon cochlear ear model. Since all the color points seem to be randomly distributed, there is no data separation. In particular, numbers in the same category will not form separate clusters. For the data set after preprocessing by SRSS, data separation can clearly be seen in Figure 9b, correlating with the word error rate of 0.2%. A few clusters such as Digit 8 and Digit 9, however, are very close to some clusters, and some even overlap with others. After optimized by the CRSD, the t-SNE result is shown in Figure 9c with WER = 0.04%. The clusters of Digit 8 and Digit 9 can be relatively well separated from the clustering of other numbers and corroborate the high recognition rates exhibited by CRSD.

The results of this isolated word classification task show that CRSD can decrease the WER by increasing its dynamic richness.

### 3.4. Human Action Recognition Task

Recognizing human actions in video streams is a difficult task in computer vision. This task was first applied to RC systems in 2019 [46] as a standard data set for testing RC systems. This recognition task is a subset of the KTH video database [47], which contains four different scenarios. For brevity, we limited the data set in the first scenario, called “S1,” which included an outdoor video of six different motions (walking, jogging, running, boxing, hand waving, and hand clapping) performed by 25 subjects. All videos were recorded with a static camera and 25 fps against a uniform background and then sampled down to 160 × 120 pixels of spatial resolution. The length of each motion is different, the average is 4 s. Our dataset consists of 25 × 6 × 4 = 600 sequences, each combination consisting of 25 subjects, 6 actions, and 4 replicates. These 600 motion sequences are preprocessed by the histograms of oriented gradients (HOG) algorithm [47], which is used to extract spatial and shape information from a single video frame. HOG features are performed by the computer, with a cell size of 8 × 8 and a block size of 2 × 2, individually for each frame of every sequence. The size of each frame is 160 × 120 pixels, and the HOG algorithm returns 19 × 14 × 4 × 9 = 9576 features per frame. These features are fed into the RC system, which outputs the values of the virtual nodes, and then trains it to classify each frame. The classifiers are implemented by defining six binary output nodes. Training them to output 1 for a frame of the corresponding class and 0 for the other classes, which is the same as training method for TI-46 isolated word classification task. Classification by frame is obtained by selecting the node with the largest output, namely the winner-take-all method. The final determination of a video sequence is given by the classes that belong to the majority of the frames in the sequence.

In this recognition task, we also use three RC systems (SRSS, SRSD, and CRSD) to test the performance, and the same parameters were set as N=3000, θ=0.1 ms, β=0.01, and $V_{dc}=20\;V$, $V_{ac}=2\;V$. The driving frequency was f=350,000 Hz in SRSS and SRSD, while f=244,000 Hz in CRSD. The delay feedback loop was α=0.2 in SRSS, while $\alpha_{1}=0.1$, $\alpha_{2}=0.1$ in SRSD and CRSD with two delay feedback loops. Figure 10 displays the confusion matrices [48] for three different RC systems. After processing by SRSS, the success rate is only 73.3%. Relatively good recognition results were obtained for the hand gestures (walking, jogging, and running), but for fast spatial movements (jogging and running) that are difficult to distinguish, the recognition results are very unsatisfactory. After optimizing the linear richness, the success rate rises to 89.33% in SRSD. The hand gestures can get perfect accuracy and the accuracy of fast spatial movements is greatly improved. The CRSD optimizes the nonlinear richness of the system, so the success rate up to 92%. Compared with SRSD, the accuracy of jogging and walking are relatively improved.

In addition, we test the effect of the number of virtual nodes N on the accuracy rate of the CRSD system. The number of virtual nodes N is a crucial parameter for RC because it determines the speed as well as the performance of the system. If N is relatively small, the performance will decrease, but the system operation speed will increase on the contrary. A higher N usually means better state diversity, which improves performance at the expense of lower computing speed. Because this recognition video data set has a much higher data complexity than the other three above, more virtual nodes are needed to process the data to increase the accuracy rate. The photonic computer gets the accuracy rate = 91.3% at the condition of N=16,384 [46]. In our test, the accuracy rate as a function of the number of virtual nodes is shown in Figure 11. In the range of N=400 to N=3000, as the number of virtual nodes increases, the accuracy rate increases significantly. When N>3000, the accuracy rate fluctuates in a small range and gradually stabilizes. The simulation time increases rapidly as the number of virtual nodes increases. If increasing the number of virtual nodes does not increase the accuracy rate, then too large a number of virtual nodes will seriously affect the calculation efficiency. As a compromise between the performance and the efficiency, we choose *N* = 3000 in this task.

## 4. Conclusions

We propose the coupled MEMS resonators can optimize the performance of the RC system based on the enhanced dynamic richness that is composed of linear richness and nonlinear richness. In order to set forth this point, we compare three typical RC structures with a single-nonlinearity RC structure with single-feedback, a single-nonlinearity RC structure with double-feedbacks that is treated as a transition structure, and the couple-nonlinearity RC structure with double-feedbacks. We also describe the evolution for the promotion of the performance of the three structures. At the system level, we use the delay characteristic diagram and memory capacity to illustrate that the delayed feedback loop increases the linear richness, and use the output signal timing diagram and the histogram of the virtual nodes to illustrate that the coupled resonator increases the nonlinear richness of the system. At the data set level, four different tasks, the parity benchmark task, nonlinear autoregressive moving average (NARMA) task, isolated spoken digits recognition, and human action recognition, are investigated to prove that CRSD can improve the accuracy of the system to a certain extent. These results have significant implications with respect to hardware implementation of the reservoir and open new future possibilities for exploring real-world applications of coupling resonators in the RC system.

## Figures and Tables

**Figure 1 sensors-21-02961-f001:**
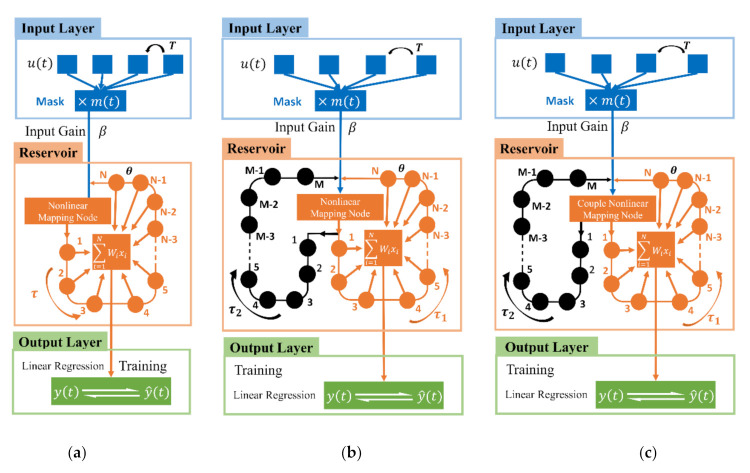
Schematic diagrams of three RC structures: (**a**) single-nonlinearity RC structure with single-feedback (SRSS); (**b**) single-nonlinearity RC structure with double-feedbacks (SRSD); and (**c**) couple-nonlinearity RC structure with double-feedbacks (CRSD). The blue part stands for the input layer and the orange part shows the reservoir layer, while the green part performs the output layer. The reservoir layer of the three structures is different, and the input layer and output layer are the same.

**Figure 2 sensors-21-02961-f002:**
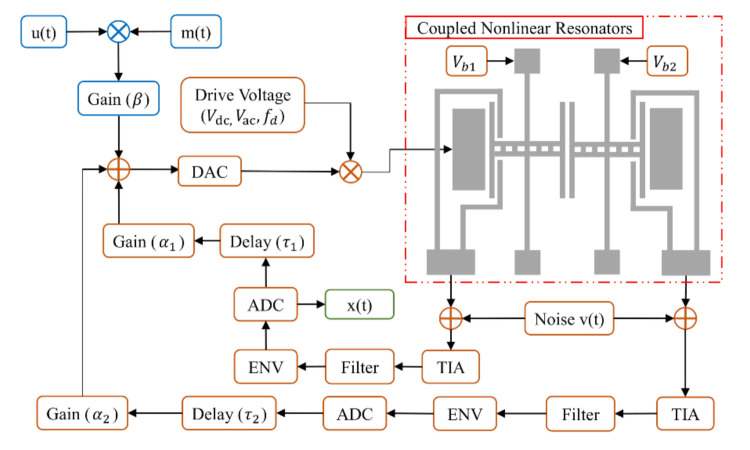
Signal chain of CRSD. The blue part stands for the input layer and the orange part shows the reservoir layer, while the green part performs the output data. The coupled MEMS resonators enclosed by the red dashed line has one signal input electrode and two signal output electrodes.

**Figure 3 sensors-21-02961-f003:**
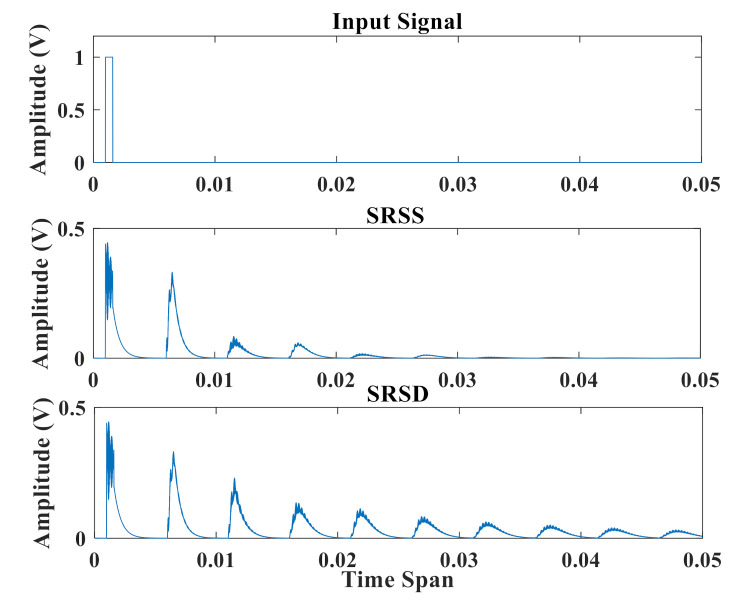
Delay characteristic diagram of two structures. From top to bottom, respectively, waveforms show the input signal, the output signal of SRSS, and the output signal of SRSD.

**Figure 4 sensors-21-02961-f004:**
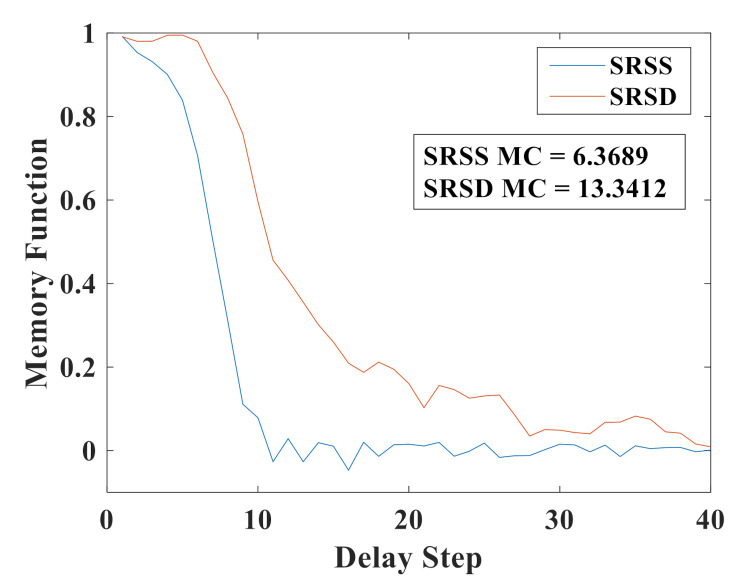
Memory curve for two RC structures. The blue line stands for the SRSS and the red line represents the SRSD.

**Figure 5 sensors-21-02961-f005:**
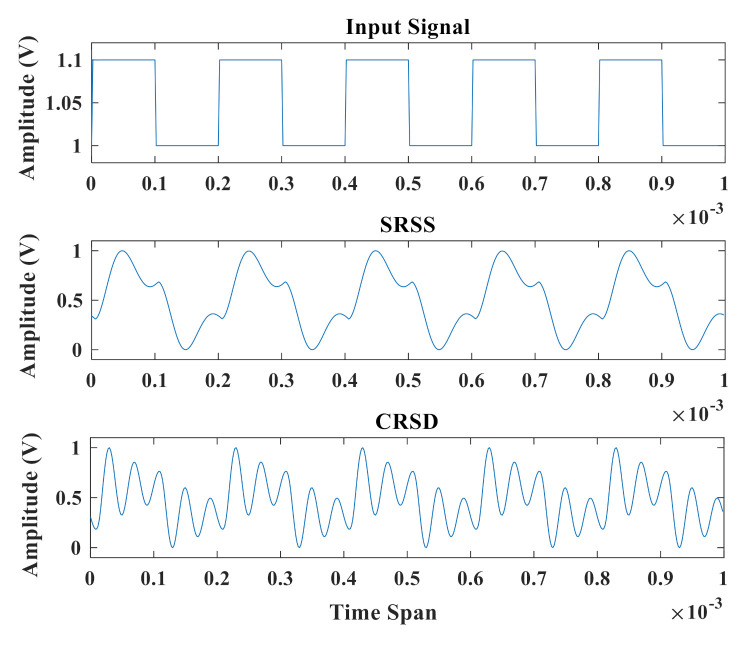
Output signal timing diagram of two reservoir layer structures. From top to bottom, respectively, waveforms show the input signal, the output signal of SRSS without the feedback loop, and the output signal of CRSD without the feedback loop.

**Figure 6 sensors-21-02961-f006:**
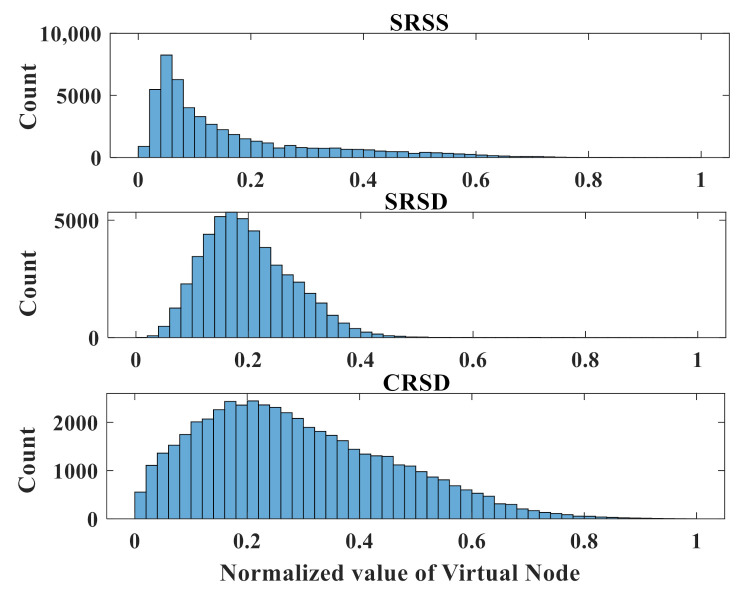
The histogram of the virtual nodes in three RC structures. From top to bottom, the histogram shows the virtual nodes of SRSS, SRSD, CRSD. For the convenience of comparison, the values of virtual nodes are all normalized.

**Figure 7 sensors-21-02961-f007:**
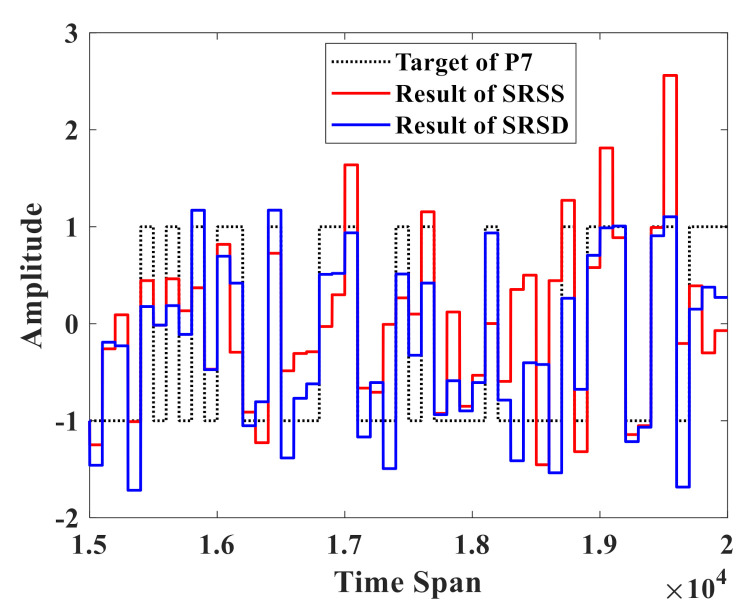
Performance of two RC systems for the parity benchmark test with n=7. The black dotted line stands for the target of $P_{7}$ and the red line represents the result of SRSS, while the blue line shows the result of SRSD in the time span of ($1.5\times 10^{4}$, $2\times 10^{4}$) after the training phase. The success rate is 78% for SRSS and 100% for SRSD.

**Figure 8 sensors-21-02961-f008:**
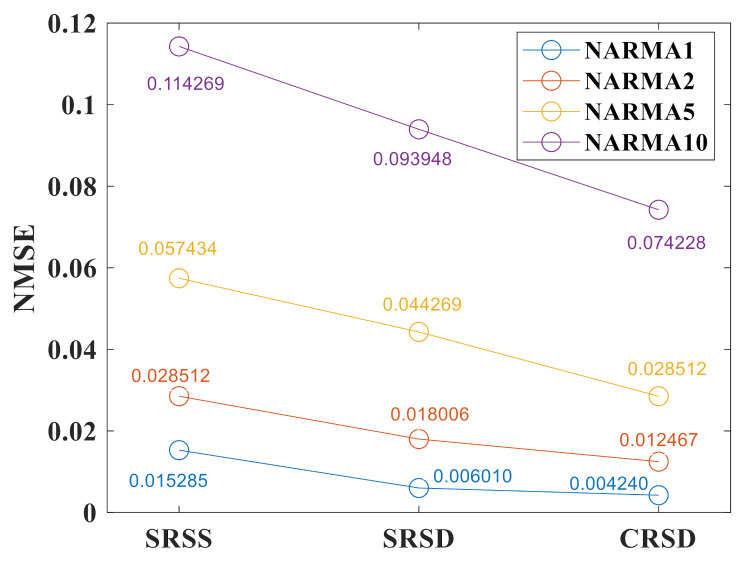
Performance of three RC systems for the NARMA prediction test with m=1,2,5,10. The blue line stands for the NMSE result of NARMA1, the orange line represents the result of NARMA2, and the yellow line represents the result of NARMA5, while the purple line shows the result of NARMA10 for three different RC systems. The numbers on the circle represent the results of the NMSE.

**Figure 9 sensors-21-02961-f009:**
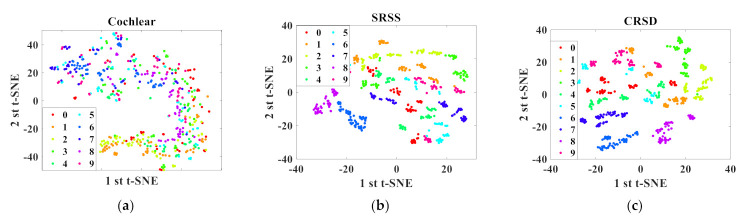
Two-dimensional representation of the two t-SNE components for SRSS and CRSD: (**a**) t-SNE result of TI-46. raw data only after processing by the Lyon cochlear ear model; (**b**) t-SNE result of TI-46 data after preprocessing by SRSS; and (**c**) t-SNE result of TI-46 data after preprocessing by CRSD.

**Figure 10 sensors-21-02961-f010:**
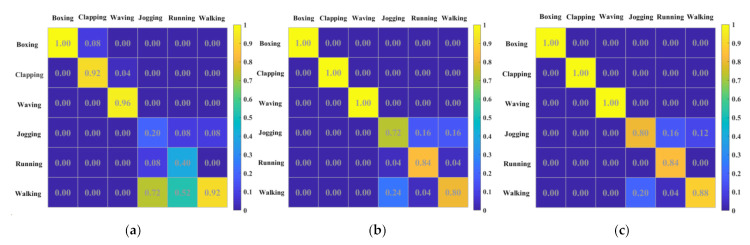
The confusion matrixes of the human action recognition benchmark show the probability (colors) that a motion presented to the system (columns) is classified to a certain motion (lines). (**a**) Confusion matrices for the SRSS with the success rate = 73.3% (**b**) Confusion matrices for the SRSD with the success rate = 89.33%. (**c**) Confusion matrices for the CRSD with the success rate = 92%.

**Figure 11 sensors-21-02961-f011:**
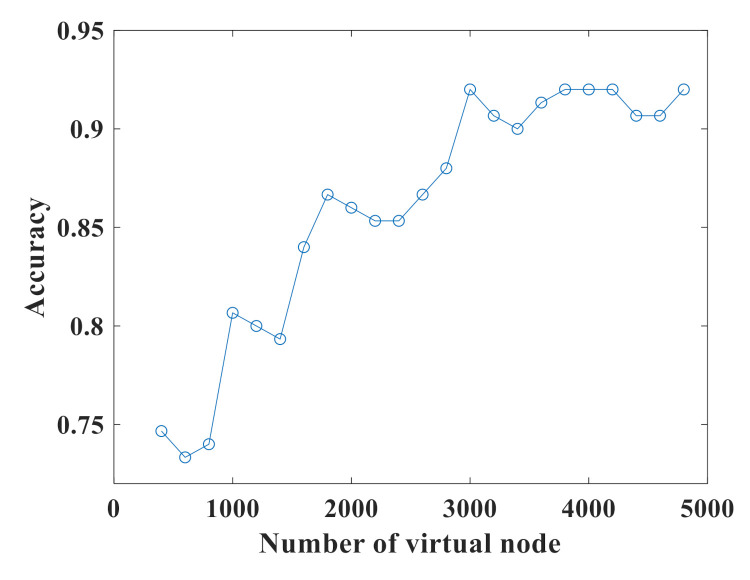
Virtual node sweep results of the human action task.

## Data Availability

Data are contained within the article. More detailed data and data presented in this study are available on request from the corresponding author. Part of the data could be included in the final reports to the corresponding funding organizations.

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
