# Peer review of "Enhancing Performance of Reservoir Computing System Based on Coupled MEMS Resonators"

_sensors, 2021, doi:10.3390/s21092961_

Round 1

Reviewer 1 Report

This manuscript presents a novel Reservoir Computing (RC) methodology in the fast growing area of Artificial Neural Networks. The RC methodology is a competing framework of the Attention Networks based on the Long Short Term Memory (LSTM) framework. Specifically RC offers a much faster learning approach compared to Attention networks at the expense of accuracy.

Overall the manuscript is well written and the research comprehensive. The importance of this work lies with the fact that the authors deal with a dynamic problem and learning of dynamic features, as opposed to traditionally static or steady problems. This dynamic feature makes the whole process significantly more challenging. 

The methodology seems sound and I very much appreciated seeing the confusion matrices in Fig. 10. This is critical to the assessment of the effectiveness of the approach. In that respect the authors should cite the following recent articles published in Nature Scientific Reports:

Lakkam et al. (2019). Hydrodynamic object identification with artificial neural models. Scientific reports, 9(1), 1-12.

Also the authors should clarify the results about the virtual node sweep shown in Fig. 11. I have had troubles understanding what exactly was going on.

Lastly, the 2D representation shown in Figure 9 would benefit from using larger figures.

Reviewer 2 Report

This is very interesting work on utilizing nonlinear coupled MEMS resonators for Reservoir Computing in artificial intelligence applications. The work is of good quality as well as the presentation. Some comments below:

- Please be more specific on if your work presented is based on modeling/simulations or if you built a physical system. If FEM simulations on the MEMS devices are included, please give more details as well.

- Are there other work related to using resonators or nonlinear resonators (either electromechanical or electromagnetic/LC resonators) that utilize similar concepts and should be cited in the intro? I found the following papers, please judge to see if these are appropriate for reference discussions.

[1] B. Barazani, et al., “Microfabricated Neuroaccelerometer: Integrating Sensing and Reservoir Computing in MEMS,” Journal of Microelectromechanical Systems, vol. 29, no. 3, June 2020.

[2] F. Denis-Le Coarer, et al., “All-Optical Reservoir Computing on a Photonic Chip Using Silicon-Based Ring Resonators,” IEEE Journal of Selected Topics in Quantum Electronics, vol. 24, no. 6, November/December 2018.

[3] L. C. G. Govia, et al., “Quantum reservoir computing with a single nonlinear oscillator,” Physical Review Research, vol. 3, 013077, 2021.

- Does the resonant frequency of MEMS devices limit the speed of computing/operation, since MEMS frequencies should be lower than electromagnetic resonators?

- Equation 2(a) and 2(b) do seem to have more nonlinear dynamic richness, but the highest nonlinear term is still third power. Compared to equation (1), does this really improve the performance?

Round 2

Reviewer 2 Report

I have no more comments.